# Regenerative Peripheral Nerve Interfaces (RPNIs) in Animal Models and Their Applications: A Systematic Review

**DOI:** 10.3390/ijms25021141

**Published:** 2024-01-17

**Authors:** Jorge González-Prieto, Lara Cristóbal, Mario Arenillas, Romano Giannetti, José Daniel Muñoz Frías, Eduardo Alonso Rivas, Elisa Sanz Barbero, Ana Gutiérrez-Pecharromán, Francisco Díaz Montero, Andrés A. Maldonado

**Affiliations:** 1Peripheral Nerve Unit, Department of Plastic Surgery, University Hospital of Getafe, 28905 Madrid, Spain; jorgegonzalezprieto240296@gmail.com (J.G.-P.); dralaracristobal@gmail.com (L.C.); 2Department of Medicine, Faculty of Biomedical Science and Health, Universidad Europea de Madrid, 28670 Madrid, Spain; 3Animal Medicine and Surgery Department, Complutense University of Madrid, 28040 Madrid, Spain; vetuihug@yahoo.es; 4Institute for Research in Technology, ICAI School of Engineering, Comillas Pontifical University, 28015 Madrid, Spain; romano@comillas.edu (R.G.); daniel@icai.comillas.edu (J.D.M.F.);; 5Peripheral Nerve Unit, Neurophysiology Department, University Hospital of Getafe, 28905 Madrid, Spain; esanzbb@gmail.com; 6Peripheral Nerve Unit, Pathological Anatomy Department, University Hospital of Getafe, 28905 Madrid, Spain; agpecharroman@salud.madrid.org; 7Department of Design, BAU College of Arts & Design of Barcelona, 28036 Barcelona, Spain; info@autofabricantes.org

**Keywords:** Regenerative Peripheral Nerve Interfaces (RPNIs), animal models, Inlay-RPNI, Burrito-RPNI, neuroma prevention, myoelectric prostheses, systematic review

## Abstract

Regenerative Peripheral Nerve Interfaces (RPNIs) encompass neurotized muscle grafts employed for the purpose of amplifying peripheral nerve electrical signaling. The aim of this investigation was to undertake an analysis of the extant literature concerning animal models utilized in the context of RPNIs. A systematic review of the literature of RPNI techniques in animal models was performed in line with the PRISMA statement using the MEDLINE/PubMed and Embase databases from January 1970 to September 2023. Within the compilation of one hundred and four articles employing the RPNI technique, a subset of thirty-five were conducted using animal models across six distinct institutions. The majority (91%) of these studies were performed on murine models, while the remaining (9%) were conducted employing macaque models. The most frequently employed anatomical components in the construction of the RPNIs were the common peroneal nerve and the extensor digitorum longus (EDL) muscle. Through various histological techniques, robust neoangiogenesis and axonal regeneration were evidenced. Functionally, the RPNIs demonstrated the capability to discern, record, and amplify action potentials, a competence that exhibited commendable long-term stability. Different RPNI animal models have been replicated across different studies. Histological, neurophysiological, and functional analyses are summarized to be used in future studies.

## 1. Introduction

Regenerative Peripheral Nerve Interfaces (RPNIs) represent a groundbreaking approach at the intersection of biomedical engineering, neurology, and regenerative medicine. These interfaces have the potential to revolutionize the field of bionic prostheses by facilitating communication between the nervous system and external devices and deterring the development of neuromas [1,2,3,4,5,6,7]. Unlike traditional neural interfaces, which rely on electrodes implanted into nerves, RPNIs aim to create a more seamless connection by harnessing the regenerative capacity of peripheral nerves [8].

RPNIs involve surgically grafting a small segment of a patient’s muscle, typically from an area with low functional significance, to a region near a damaged nerve or a residual limb [6,9,10,11,12,13]. The regenerative nature of peripheral nerves allows the nerve to regrow, reinnervate, and revascularize the muscle graft in three to four months [6,9,14,15,16,17,18]. This results in the formation of a “bioelectrode” that can be utilized to transmit signals between the patient’s nervous system and prosthetic devices [10,19,20,21,22].

Animal models are integral to advancing our understanding of RPNIs in biomedical research. By replicating and studying the RPNI murine model across various contexts, researchers gain insights into its mechanisms and potential applications in neuroprosthetics and pain management [1,9,10,11,13,15,23,24,25,26,27,28,29,30,31,32,33,34,35,36,37,38,39,40,41,42,43,44,45,46,47,48]. These models help to assess reproducibility, long-term viability, and functional changes in RPNI constructs. This knowledge is crucial for translating RPNI innovations into effective treatments for limb impairments and neurological conditions [5,12,49,50,51,52,53,54,55,56,57,58,59].

The aim of this study was to perform the first systematic literature review of the RPNI technique across animal models. The different applications and characteristics of each model are analyzed. We believe the knowledge of all of the different surgical techniques and the different histological, neurophysiological, and functional tests may be useful for future research projects involving RPNIs.

## 2. Materials and Methods

### 2.1. Search Strategy

A comprehensive literature review was executed by searching the MEDLINE/PubMed and Embase databases spanning from 1 January 1970, to 30 September 2023. The search process encompassed both automated and manual approaches, ensuring the identification of all pertinent literature. The adherence to the PRISMA statement (Preferred Reporting Items for Systematic Reviews and Meta-Analysis) [60] guided the execution and reporting of this review. Employing English keywords along with Boolean logical operators, specifically “(RPNI) OR (Regenerative Peripheral Nerve Interfaces)”, facilitated the search process. Notably, no limitations were imposed during the search.

### 2.2. Selection Criteria (Figure 1)

We included articles written in either English or Spanish that either described or employed the RPNI technique. Exclusion criteria encompassed studies where the RPNI technique was not utilized or was applied in non-animal models. We omitted duplicated studies and articles from the same author or author groups if they were identical. The evaluation of titles, abstracts, and full text, as well as the application of inclusion and exclusion criteria, was carried out independently by two independent plastic surgeons (J.G. and A.A.M.). Full versions of potentially relevant studies were procured for further assessments. Additional articles were considered following a review of the references from the retrieved articles. In cases of disagreement between the two reviewers, resolution was achieved through discussion and consensus.

**Figure 1 ijms-25-01141-f001:**
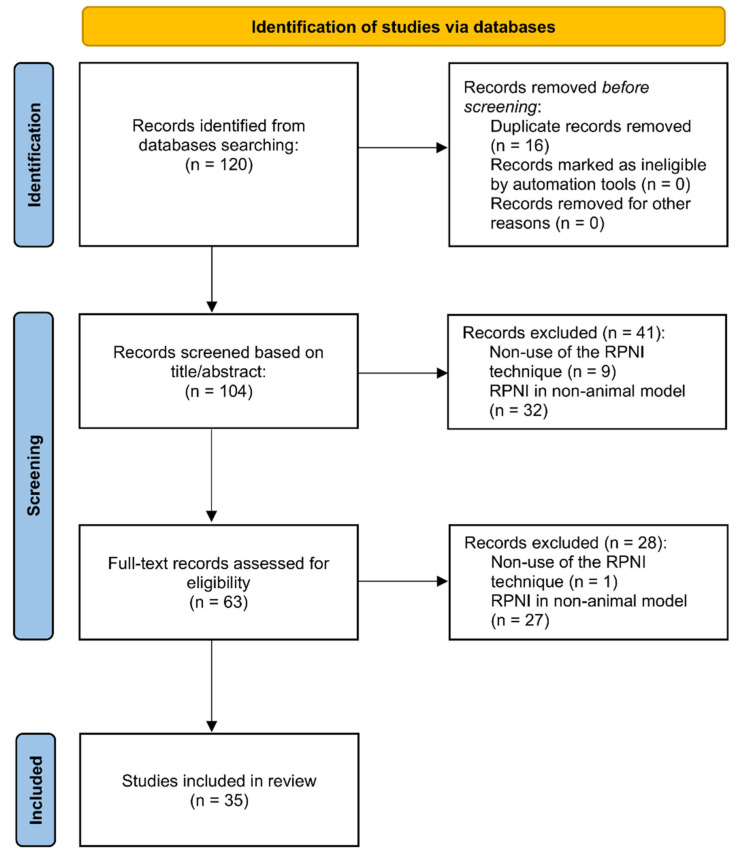
PRISMA flow diagram.

### 2.3. Data Extraction

The data were gathered using the software “Microsoft Excel for Mac, version 16.8 (23121017)”. The collected data encompassed various elements: the article database, the university center where the study took place, the publication year, the study’s objectives and categorized groups, the animal species, along with the total animal count used, specifics about the nerve and muscle utilized in constructing the RPNI, the design and model of RPNI construction, the selection of histological parameters (including muscle angiogenesis, tissue viability, muscle axonal regeneration, neuroma formation, and fibrosis/scarring), a subset of functional variables (encompassing stimulus intensity and localization, compound muscle action potential (CMAP), motor unit action potential (MUAP), compound sensory nerve action potential (CSNAP) measurements, latency periods, and maximum muscle force), the outcomes, and the average follow-up duration.

A comprehensive quantitative analysis of the quality and limitations of the selected studies was conducted. This process was carried out following the “10 Essential ARRIVE (Animal Research: Reporting In Vivo Experiments)” outlined in the guidelines of “ARRIVE guidelines 2.0” [61]. Each study was rated on a 10-point scale, considering the degree of compliance with the evaluated items. These criteria represent the minimum requirement of information necessary to ensure that reviewers and readers can assess the reliability of the presented findings.

## 3. Results

In the initial search results, one hundred and twenty articles were identified through manual searching. These one hundred and twenty articles underwent an initial screening process, during which sixteen were eliminated due to duplication. Additionally, forty-one articles were excluded based on their title or abstract, and twenty-eight more were excluded after a thorough examination of the full text. Among these exclusions, ten articles were discarded because they did not employ the RPNI technique, and fifty-nine were excluded as they did not involve an animal model of RPNI. As a result, the final review comprises thirty-five articles (Figure 1).

Among the thirty-five articles under consideration, thirty have been disseminated via MEDLINE/PubMed [1,3,9,10,11,13,15,23,24,25,26,27,28,29,30,31,32,33,34,35,36,37,38,39,40,41,46,47,48,62], while five have found their place in the Embase database [42,43,44,45,63].

The University of Michigan emerges as the foremost research institution in the domain of the RPNI technique employed in animal models, with a noteworthy presence in twenty-eight (80%) publications. In twenty-four instances, it stands alone as the primary research center, while in four instances, it collaborates with the universities of Alberta, British Columbia, Delaware, and Groningen. The remaining seven (20%) articles originate from the universities of Wuhan, Beijing, Florida, Cambridge, and Texas (Table 1).

**Table 1 ijms-25-01141-t001:** General data of studies.

Reference No.	PMID	First Author	Database	University	Publication Year	Study Groups	Follow-Up Time (Months)	Quality Score
[23]	36729137	Ian C. Sando	PubMed	Michigan	2022	1—Control Full-thickness Skin (CFS), 2—Control De-epithelialized Skin (CDS), 3—Control Transected Nerve (CTN), 4—Dermal Sensory Interface (DS-RPNII)	5	9
[24]	34359056	Carrie A. Kubiak	PubMed	Michigan	2021	1—8 mm MC-RPNI with epineural window, 2—8 mm MC-RPNI without epineural window, 3—13 mm MC-RPNI with epineural window, 4—13 mm MC-RPNI without epineural window	3	10
[15]	32176203	Shelby R. Svientek	PubMed	Michigan	2021	1—C-RPNI (compound regenerative peripheral nerve interface)	9	6
[25]	36161173	Zheng Wang	PubMed	Wuhan	2022	1—Control, 2—NSR (nerve stump implantation inside a fully innervated muscle), 3—RPNI	1.5	10
[26]	35875668	Jiaqing Wu	PubMed	Beijing	2022	1—Control, 2—RPNI	2	10
[11]	30458876	Christopher M. Frost	PubMed	Michigan	2018	1—Control, 2—Denervated, 3—RPNI	5	9
[27]	26859115	Daniel Ursu	PubMed	Michigan	2016	1—Control, 2—Denervated, 3—RPNI	4	8
[28]	28438166	Daniel Ursu	PubMed	Michigan	2017	1—Control, 2—RPNI	4	8
[29]	25569986	Christopher M. Frost	PubMed	Michigan/IEEE	2014	1—NerveStim, 2—DirectStim, 3—DirectSIS, 4—DirectPEDOT	0	9
[30]	25570372	Nicholas B. Langhals	PubMed	Michigan/IEEE	2014	1—RPNI	14	7
[62]	27247270	Zachary T. Irwin	PubMed	Michigan	2016	1—RPNI	20	7
[13]	33290586	Yaxi Hu	PubMed	Michigan and Groningen	2020	1—RPNI 150 mg, 2—RPNI 300 mg, 3—RPNI 600 mg, 4—RPNI 1200 mg, 5—Control	3	9
[1]	27294122	Melanie G. Urbanchek	PubMed	Michigan and Delaware	2016	1—Silicone mesh, 2—Acellular muscle, 3—Acellular muscle with a conductive polymer (PEDOT)	3	7
[31]	35098950	Shelby R. Svientek	PubMed	Michigan	2022	1—MC-RPNI, 2—Control	3	7
[3]	29432117	Philip P. Vu	PubMed	Michigan/IEEE	2018	1—RPNI, 2—Control (ECR)	12	9
[32]	25570963	Shoshana L. Woo	PubMed	Michigan/IEEE	2014	1—RPNI, 2—RPNI with Tibial anterior (TA) and Extensor hallucis longus (EHL) muscles excision	1.5	8
[9]	24867721	Theodore A. Kung	PubMed	Michigan	2013	1—RPNI with steel electrode, 2—RPNI + PEDOT, 3—Control with steel electrode, 4—RPNI + PEDOT	7	8
[33]	25942171	Andrej Nedic	PubMed	Michigan	2014	1—Control, 2—RPNI, 3—Denevated	Not specified	8
[34]	25942129	Christopher M. Frost	PubMed	Michigan	2014	1—Control, 2—RPNI, 3—Denervated	5	8
[35]	25942128	John V Larson	PubMed	Michigan	2014	1—Control, 2—RPNI	4	8
[36]	22456363	Christopher M. Frost	PubMed	Michigan	2012	1—RPNI with steel electrode, 2—RPNI + PEDOT	Not specified	10
[10]	26502083	Ian C. Sando	PubMed	Michigan	2016	1—Epimysial electrode + bipolar/monopolar stimulation, 2—Intramuscular electrode + bipolar/monopolar stimulation	4	8
[37]	32413377	Benjamin S. Spearman	PubMed	University of Florida	2020	1—RPNI TEENI	Not specified	6
[38]	36204848	Zheng Wang	PubMed	Wuhan	2022	1—RPNI, 2—NSM (nerve stump implantation inside a fully innervated muscle), 3—Denervated/control	2	10
[39]	25942172	Zachary P French	PubMed	Michigan	2014	1—Control, 2—RPNI	5	8
[40]	35998559	Eric W Atkinson	PubMed	University of Florida	2022	1—RPNI MARTEENI	2.5	8
[41]	19744916	Stéphanie P. Lacour	PubMed	Cambridge	2009	1—Group 1, 2—Group 2, 3—Group 3	3	8
[63]	L71587711	Zachary T. Irwin	Embase	Michigan	2014	1—RPNI	6	5
[42]	L71676463	Shoshana Woo	Embase	Michigan	2014	1—Extensor digitorum longus (EDL), 2—Biceps femoris, 3—Rectus femoris, 4—Gastrocnemius, 5—Vastus medialis	4	8
[43]	L71254630	Ziya Baghmanli	Embase	Michigan	2011	1—Exposed soleus muscle not transferred or neurotized, 2—Soleus muscle transferred and neurotized, 3—Soleus muscle transferred, but not neurotized	1	5
[45]	L71606159	Ian C Sando	Embase	Michigan	2014	1—Control, 2—RPNI	5	6
[44]	L71587616	Bongkyun Kim	Embase	Texas	2014	1- RPNI	Not specified	5
[46]	37265342	Jenna-Lynn B. Senger	PubMed	Michigan, Alberta and British Columbia	2023	1—Target muscle reinnervation (TMR), 2—RPNI, 3—Neuroma excision, 4—Neuroma in situ	1.5	10
[47]	37227138	Amir Dehdashtian	PubMed	Michigan	2023	1—Neuroma, 2—RPNI, 3—Control	2	9
[48]	37400949	Jenna-Lynn Senger	PubMed	Michigan, Alberta and British Columbia	2023	1—Inlay-RPNI, 2—Burrito-RPNI, 3—Control	4	9

The average duration of follow-up for the animal subjects spanned 4.7 months (range, 0–20 months). The longest-running research extended for twenty months [62].

With the purpose of assessing the quality of the included research, a comprehensive quantitative analysis of the 35 studies was conducted. The evaluation resulted in a final average score of 8 out of 10, demonstrating a strong level of compliance with minimum reporting standards.

### 3.1. Type of Models (Table 2)

#### 3.1.1. Species Selection and Sample Size

The selection of animal species for research purposes predominantly featured rats in thirty-two (91%) instances [1,9,10,11,13,15,23,24,25,26,27,28,29,30,31,32,33,34,35,36,37,38,39,40,41,42,43,44,45,46,47,48], with Rhesus macaques employed on the remaining three occasions [3,62,63]. The sample size across these investigations exhibits a range from 2 to 90 specimens, averaging 20 animals per study.

**Table 2 ijms-25-01141-t002:** RPNI models and components of studies.

Reference No.	Animal	No. of Animals	RPNI Design	Nerve	Muscle	RPNI Model	Aim
[23]	Rat	40	Burrito RPNI	Sural	Not specified	Sensible (DS-RPNI)	HA, NA
[24]	Rat	37	Burrito RPNI (nerve in-continuity)	Common peroneal	EDL (Extensor digitorum longus)	Motor (MC-RPNI)	HA, NA
[15]	Rat	Not specified	Inlay RPNI	Common peroneal	EDL	Mixed (C-RPNI)	HA, NA
[25]	Rat	60	Inlay RPNI	Sciatic	Adductor magnus	Motor	NP
[26]	Rat	22	Burrito RPNI	Sciatic	EDL	Motor	NP, HA
[11]	Rat	6	Inlay RPNI	Common peroneal	EDL	Motor	MP, NA
[27]	Rat	6	Inlay RPNI	Peroneal and tibial	EDL	Motor	NA
[28]	Rat	4	Inlay RPNI	Common peroneal and tibial	EDL	Motor	NA
[29]	Rat	5	Not specified	Common peroneal	EDL	Motor	NA
[30]	Rat	Not specified	Inlay RPNI	Common peroneal	EDL	Motor	HA, NA
[62]	Rhesus macaque	2 (9 RPNIs in total)	Burrito RPNI	Median and radial	Flexor carpi radialis (FCR), flexor digitorum superficialis (FDS), and extensor digitorum communis (EDC)	Motor	MP, HA, NA
[13]	Rat	30	Inlay RPNI	Common peroneal	Semimembranosus	Motor	HA, NA
[1]	Rat	25	Burrito RPNI	Common peroneal	Soleus	Motor	NP, HA
[31]	Rat	12	Burrito RPNI (without nerve section)	Common peroneal	EDL	Motor	HA, NA
[3]	Rhesus macaque	2 (7 RPNIs in total)	Burrito RPNI	Median and radial	Flexor digitorum profundus (FDP), FDS, and EDC	Motor	MP, NA
[32]	Rat	18	Inlay RPNI	Common peroneal	EDL	Motor	NA
[9]	Rat	16	Inlay RPNI	Common peroneal	EDL	Motor	HA, NA
[33]	Rat	9	Not specified	Tibial	Soleus	Motor	NA
[34]	Rat	6	Not specified	Tibial	Soleus	Motor	MP, NA
[35]	Rat	12	Not specified	Sural	EDL	Motor	HA, NA
[36]	Rat	18	Inlay RPNI	Common peroneal	EDL	Motor	NA
[10]	Rat	8	Not specified	Common peroneal and tibial	EDL	Motor	MP, HA, NA
[37]	Rat	3	Not specified	Sciatic	Not specified	Motor	NA
[38]	Rat	90	Burrito RPNI	Sciatic	Adductor magnus	Motor	NP, HA
[39]	Rat	10	Not specified	Common peroneal	EDL	Motor	NA
[40]	Rat	5	Not specified	Sciatic	Not specified	Motor	NA
[41]	Rat	30	Not specified	Sciatic	Not specified	Motor	HA
[63]	Rhesus macaque	Not specified	Not specified	Median	FDS, FDP, and flexor pollicis longus (FPL)	Motor	MP, NA
[42]	Rat	20	Not specified	Common peroneal	EDL, biceps femoris, rectus femoris, gastrocnemius, and vastus medialis	Motor	NA
[43]	Rat	Not specified	Not specified	Tibial	Soleus	Motor	NA
[45]	Rat	5	Not specified	Common peroneal	EDL	Motor	NA
[44]	Rat	Not specified	Not specified	Sciatic	Tibialis anterior, soleus, and vastus lateralis	Motor	NA
[46]	Rat	36	Inlay RPNI	Tibial	EDL and biceps femoris	Motor	NP, HA
[47]	Rat	36	Inlay RPNI	Tibial	EDL	Motor	NP, HA
[48]	Rat	18	Inlay RPNI and Burrito RPNI	Tibial	EDL	Motor	NP, HA

NP = Neuroma prevention, MP = Myoelectric prostheses, HA = Histological analysis, NA = Neurophysiological analysis.

#### 3.1.2. RPNI Construction Designs

Two distinct designs for RPNI construction are prevalent among the studies: the Inlay-RPNI (in which the nerve is inset into the muscle graft and secured within an intact muscle belly) in thirteen (37%) articles [9,11,13,15,25,27,28,30,32,36,46,47,48] and the Burrito-RPNI (in which the muscle graft is wrapped around the distal nerve stump) in nine (26%) articles [1,3,23,24,26,31,38,48,62]. However, fourteen articles do not specify the particular model employed [10,29,33,34,35,37,39,40,41,42,43,44,45,63].

A different type of Burrito RPNI construction has been published [31] using a segment of a free muscle graft wrapping an intact peripheral nerve (the muscle is placed above the epineurium of a nerve, which is not sectioned).

#### 3.1.3. Nerve and Muscle Selection

In constructing the RPNI, the common peroneal nerve takes precedence in seventeen (49%) cases [1,9,10,11,13,15,24,27,28,29,30,31,32,36,39,42,45], and in three of these, it is combined with the tibial nerve [10,27,28]. The tibial nerve is employed in nine (26%) studies [10,27,28,33,34,43,46,47,48]. Less frequently used nerves include the sciatic nerve in seven (20%) studies [25,26,37,38,40,41,44], the sural nerve in two (6%) studies [23,35], the median nerve in three (9%) studies [3,62,63] and the radial nerve in two (6%) instances [3,62], both of them in conjunction with the median nerve.

The extensor digitorum longus (EDL) muscle emerges as the preferred choice for RPNI construction in twenty (57%) instances [9,10,11,15,24,26,27,28,29,30,31,32,35,36,39,42,45,46,47,48]. Other less frequently selected muscles include the soleus muscle in five (14%) studies [1,33,34,43,44], the flexor digitorum superficialis (FDS) in three (9%) [3,62,63], and the adductor magnus [25,38], extensor digitorum communis (EDC) [3,62], flexor digitorum profundus (FDP) [3,63], and biceps femoris [42,46] in two (6%) studies each. Additionally, the flexor pollicis longus (FPL) [63], flexor carpi radialis (FCR) [62], semimembranosus [13], rectus femoris [42], gastrocnemius [42], vastus medialis [42], vastus lateralis [44], and tibialis anterior [44] are employed in one study each. Four articles omit the specification regarding the muscle utilized for RPNI construction [23,37,40,41].

#### 3.1.4. Motor vs. Sensory Model

In the realm of RPNI animal models, the motor model is featured in thirty-three (94%) articles [1,3,9,10,11,13,24,25,26,27,28,29,30,31,32,33,34,35,36,37,38,39,40,41,42,43,44,45,46,47,48,62,63], the sensory model (DS-RPNI) in a single study [23], and the mixed model (C-RPNI) in another study [15].

### 3.2. Aim of the Study

We found four main types of studies regarding the research question:Aim 1: Neuroma prevention (NP) [1,25,38,46,47,48]. These groups of six (17%) publications seek to assess the efficacy of RPNI in preventing the development of neuromas and alleviating neuropathic pain.Aim 2: Myoelectric prostheses development (MP) [3,10,11,34,62,63]. This set of six (17%) studies focuses on examining and evaluating RPNI for its possible use within myoelectric prosthetic devices.Aim 3: Histological analysis (HA) [1,9,10,13,15,23,24,26,30,31,35,38,41,46,47,48,62]. This series of seventeen (49%) articles primarily concentrates on assessing the muscle viability associated with RPNI construction.Aim 4: Neurophysiological analysis (NA) [3,9,10,11,13,15,23,24,27,28,29,30,31,32,33,34,35,36,37,39,40,42,43,44,45,62,63]. This research group of twenty-seven (77%) articles is dedicated to enhancing and streamlining the acquisition and amplification of electrical signals from the RPNI muscle, aiming to optimize their application.

### 3.3. Histological Analysis (Table 3)

Histological analysis was performed in eighteen studies. The assessment of tissue viability in RPNIs yielded satisfactory results in all of these eighteen articles [1,9,10,13,15,23,24,26,30,31,35,38,41,44,46,47,48,62]. It has been noted that tissue viability diminishes proportionally with an increasing muscle graft mass surpassing 300 mg [13]. Analytical methodologies encompass direct visual examination through electron microscopy, staining procedures using hematoxylin-eosin, Masson’s trichrome, and the deployment of anti-desmin monoclonal antibodies (D33), as well as comparisons between the initial and final muscle graft weights and an evaluation of muscular response to electrical nerve stimulation.

The evaluation of muscular neoangiogenesis in RPNIs reveals robust outcomes in the fourteen studies where it was assessed through a histopathological analysis [1,9,10,13,15,23,24,30,31,35,41,46,47,62]. The muscular neoangiogenesis deteriorates in direct correlation with an increase in the muscle graft mass exceeding 300 mg [13]. Various techniques employed for analysis encompass direct visual examination via electron microscopy, staining techniques, such as hematoxylin-eosin, Masson’s trichrome, and DAPI (4′,6-diamidino-2-phenylindole) protocols, and the application of anti-Pzero and anti-RECA1 monoclonal antibodies.

Axonal regeneration was confirmed in the sixteen articles that undertook a histopathological analysis in this regard [1,9,13,15,23,24,26,30,31,35,40,41,44,46,47,62]. A noteworthy observation underscores that axonal regeneration deteriorates with an increase in the muscle graft mass beyond 300 mg [13]. Several techniques applied for analysis include direct visual inspection through electron microscopy and staining techniques incorporating hematoxylin-eosin, Toluidine blue, and acetylcholinesterase, as well as the DAPI protocol, anti-filament antibodies, anti-alpha bungarotoxin, anti-neurofilament 200 (NF 200), anti-neurofilament H (NF H), anti-neurofilament S100 (NF S100), anti-Pzero, and anti-RECA1 antibodies. Additionally, the muscle response to nerve electrical stimulation contributes to the assessment.

Neuroma formation within the RPNI was evaluated in twelve articles [1,9,13,15,23,24,26,31,38,46,47,48]. Generally, neuroma formation was not evident in the RPNI; however, it has been observed to increase proportionally with an escalation in the muscle graft mass exceeding 300 mg [13]. The rate of neuroma formation is higher when employing the targeted muscle reinnervation (TMR) technique as opposed to RPNI and with the Burrito-RPNI in comparison to the Inlay-RPNI [46,48]. Analytical techniques encompass direct visual examination through electron microscopy and staining procedures involving hematoxylin-eosin, Toluidine blue, and Masson’s trichrome, as well as the use of anti-alpha bungarotoxin and anti-neurofilament 200 (NF200) monoclonal antibodies and ultrasonography.

Fibrosis formation within the RPNI was scrutinized in sixteen articles [1,9,10,13,15,23,24,26,30,31,35,38,40,46,47,62]. In general, fibrosis formation is not conspicuous in the RPNI; however, it has been observed to increase proportionally with an augmentation in the muscle graft mass beyond 300 mg [13]. Notably, the rate of fibrosis formation is higher when the electrode is positioned intramuscularly, but lower when it is placed epimysially [10]. Analytical techniques encompass direct visual examination via electron microscopy and staining techniques employing hematoxylin, eosin, and Masson’s trichrome, as well as the utilization of anti-alpha smooth muscle actin (α-SMA) filament monoclonal antibodies.

The array of studies encompasses other investigations, including the quantification of neuronal density, measurement of the apoptosis index via the Terminal Deoxynucleotidyl Transferase-Mediated dUTP Nick-End Labeling (TUNEL) method, measurement of marker expression (Bax, BCL-2, and NTs), and the assessment of the degree of autotomy [25,26].

**Table 3 ijms-25-01141-t003:** Histological analysis of studies.

Reference No.	Histology				
	Muscular Neoangiogenesis	Tissue Viability	Axonal Regeneration	Neuroma Formation	Fibrosis Formation
[23]	Good in all (Hematoxylin-eosin (HE) and trichrome stains)	Good in all (Hematoxylin-eosin (HE) and trichrome stains)	Yes (Anti-filament antibodies)	Small neuromas in control transected nerve group	No (Antifilament antibodies)
[24]	Good (HE stain)	Good (HE stain)	Yes (HE stain)	No (HE stain)	No (HE stain)
[15]	Good	Good	Yes	No	No
[25]	Not evaluated	Not evaluated	Not evaluated	Not evaluated	Not evaluated
[26]	Not evaluated	Good	Yes (Anti-neurofilament 200 antibodies)	Lower risk (Ultrasounds)	No in RPNI group (α-SMA immunohistochemistry)
[11]	Not evaluated	Not evaluated	Not evaluated	Not evaluated	Not evaluated
[27]	Not evaluated	Not evaluated	Not evaluated	Not evaluated	Not evaluated
[28]	Not evaluated	Not evaluated	Not evaluated	Not evaluated	Not evaluated
[29]	Not evaluated	Not evaluated	Not evaluated	Not evaluated	Not evaluated
[30]	Good (HE stain)	Good (HE stain)	No denervation data (HE stain)	Not evaluated	No (HE stain)
[62]	Good (HE stain)	Good (HE stain)	Good reinnervation (Electrical stimulation)	Not evaluated	No (HE stain)
[13]	Good in group 1 and 2 (HE stain, Masson’s trichrome and von Willebrand factor (vwf))	Good in group 1 and 2. Fibrosis and central atrophy in groups 3 and 4 (HE stain and Masson’s trichrome)	Best in group 1 and 2 (Toluidine blue)	Present in group 5 (Toluidine blue)	Fibrosis and central atrophy in groups 3 and 4 (HE and Masson’s trichrome stain).
[1]	Good (HE stain)	Good (HE stain and anti-desmin staining protocol)	Good (Acetylcholinesterase stain)	No (HE stain)	No (HE stain)
[31]	Good (HE stain)	Good (HE stain)	Yes (Anti-filament and alpha-bungarotoxin antibodies)	No (HE stain)	No (HE stain)
[3]	Not evaluated	Not evaluated	Not evaluated	Not evaluated	Not evaluated
[32]	Not evaluated	Not evaluated	Not evaluated	Not evaluated	Not evaluated
[9]	Good (Masson’s trichrome and electron microscopy)	Good (Masson’s trichrome, RPNI initial/final weight comparison, and electron microscopy)	Good (Anti-neurofilament 200 and anti-alpha-bungarotoxin antibodies)	No (Anti-neurofilament 200 and anti-alpha-bungarotoxin antibodies)	No (Masson’s trichrome and electron microscopy)
[33]	Not evaluated	Not evaluated	Not evaluated	Not evaluated	Not evaluated
[34]	Not evaluated	Not evaluated	Not evaluated	Not evaluated	Not evaluated
[35]	Good (Histomorphometric and immunohistochemical techniques)	Good (Histomorphometric and immunohistochemical techniques)	Good reinnervation (Electrical stimulation)	Not evaluated	No (Histomorphometric and immunohistochemical techniques)
[36]	Not evaluated	Not evaluated	Not evaluated	Not evaluated	Not evaluated
[10]	Good (Masson’s Trichrome)	Good (Masson’s Trichrome)	Not evaluated	Not evaluated	Fibrous capsule in group 1 and fibrosis in group 2 (Masson’s trichrome)
[37]	Not evaluated	Not evaluated	Not evaluated	Not evaluated	Not evaluated
[38]	Not evaluated	Good (Masson’s Trichrome)	Not evaluated	Present in groups 2 and 3. No neuromas in group 1 (Masson trichrome and Toluidine blue)	Lower in RPNI (Anti-α-SMA antibodies)
[39]	Not evaluated	Not evaluated	Not evaluated	Not evaluated	Not evaluated
[40]	Not evaluated	Not evaluated	Good (Anti-neurofilament H, neurofilament S100, and DNA antibodies)	Not evaluated	Fibrous capsule present around the electrode
[41]	Good (DAPI protocol, anti-Pzero antibodies, and anti-RECA1 antibodies)	Good (DAPI protocol, anti-Pzero antibodies, and anti-RECA1 antibodies)	Good (DAPI protocol, anti-Pzero antibodies, and anti-RECA1 antibodies)	Not evaluated	Not evaluated
[63]	Not evaluated	Not evaluated	Not evaluated	Not evaluated	Not evaluated
[42]	Not evaluated	Not evaluated	Not evaluated	Not evaluated	Not evaluated
[43]	Not evaluated	Not evaluated	Not evaluated	Not evaluated	Not evaluated
[45]	Not evaluated	Not evaluated	Not evaluated	Not evaluated	Not evaluated
[44]	Not evaluated	Good (Response to electrical stimulation)	Good reinnervation (Electrical stimulation)	Not evaluated	Not evaluated
[46]	Good	Good	Good (Anti-neurofilament 200 and alpha-bungarotoxin antibodies)	Lower in RPNI and TMR, although similar between both groups	Lower in RPNI
[47]	Good (HE stain)	Good (HE stain)	Good (HE stain)	No in RPNI group	No in RPNI group
[48]	Not evaluated	Good	Not evaluated	Greater in Burrito-RPNI than in Inlay-RPNI	Not evaluated

### 3.4. Neurophysiological Analysis (Table 4)

The stimulus modality included electrical stimulation in nineteen (54%) studies [9,10,13,15,23,24,29,30,31,32,35,36,39,40,42,43,44,45,62] and mechanical stimulation in nine (26%) studies [1,3,11,23,27,28,33,34,63]. Of the latter, one study employed monofilaments [11,34], three utilized a treadmill [27,28,33], and one employed nociceptive tactile stimuli [1]. In addition, in two studies conducted on macaques, voluntary finger movements served as the stimulus source [3,63].

In the case of the most prevalent model (Inlay RPNI using EDC and tibial/peroneal nerve in rats) [9,11,15,27,28,30,32,36,46,47,48], the average stimulus intensity was 49 microamperes (μA) with a range of 5 to 1500 μA. The mean CMAP was 11.45 millivolts (mV) ranging from 2.79 to 7.05 mV, the mean CSNAP measured 119.47 microvolts (μV) ranging from 104.6 to 134.34 μV, and the mean latency was 2.295 milliseconds (ms) ranging from 1.05 to 4.09 ms. The maximum muscle strength was not assessed in any of these models.

In the case of another common model (Burrito RPNI using EDC and tibial/peroneal nerve in rats) [24,31,48], the average CMAP measured 4.33 mV with a range of 0.75 to 35.3 mV, the average CSNAP was 123.3 μV with a range of 78.6 to 206.6 μV, and the mean latency was 1.175 ms with a range of 0.8 to 1.55 ms. The mean maximum muscle contraction strength was 2478.8 millinewtons (mN) with a range of 2226.7 to 2933.9 mN. The stimulus intensity was not recorded in any of these models.

When considering Rhesus macaques models [3,62,63], the average stimulus intensity in the nerve was 10.5 μA with a range of 1 to 20 μA, and in the muscle, it was 45 μA with a range of 30 to 90 μA. The average CMAP was 500 mV with a range of 400 to 600 mV. The maximum muscle strength was not evaluated in any of these models, and stimulus intensity data were not recorded for these models as well.

**Table 4 ijms-25-01141-t004:** Neurophysiological analysis of studies.

Reference No.	Principal Test	Neurophysiology				
		Stimulus Intensity	Stimulus Location	CMAP/MUP/CSNAP	Latency	Maximum Muscle Contraction Strength
[23]	Electrical and mechanical stimulation	0–800 μA	Sural nerve	Mechanical stimulation: 0.06 mV. Electrical stimulation: 0.015 to 0.04 mV	Not specified	Not specified
[24]	Electrical stimulation	Until reaching the maximum CMAP	Peroneal nerve	3.67 ± 0.58 mV to 6.04 ± 1.01 mV	Not specified	1—2341 ± 114.3, 2—2398 ± 143.7, 3—2351 ± 290.2, 4—2832 ± 101.9 mN
[15]	Electrical stimulation	Until reaching the maximum CMAP/CSNAP	Peroneal nerve	8.7 +/− 1.6 mV at 3 months and 10.2 +/− 2.1 mV at 9 months	Not specified	Not specified
[25]	Not specified	Not specified	Not specified	Not specified	Not specified	Not specified
[26]	Not specified	Not specified	Not specified	Not specified	Not specified	Not specified
[11]	Mechanical stimulation (monofilament)	Not specified	Peroneal nerve	Not specified	Not specified	Not specified
[27]	Mechanical stimulation (treadmill at 8.5–9 m/min)	Not specified	Peroneal and tibial nerve	0.75 to 1.0 mV during walking and <0.1 mV during rest	Not specified	Not specified
[28]	Mechanical stimulation (treadmill at 8.5–9 m/min)	Not specified	Peroneal and tibial nerve	0.75 to 1.0 mV during running	Not specified	Not specified
[29]	Electrical stimulation	Until reaching the maximum muscle contraction strength	Peroneal nerve and EDL	Not specified	Not specified	The maximum specific muscle force was statistically greater in group 1 than group 2
[30]	Electrical stimulation	400–1500 μA	Peroneal nerve and EDL	Until >4 mV	Not specified	Not specified
[62]	Electrical stimulation	1000–20,000 μA in nerve y 30,000–60,000 μA in muscle	FCR, FDS and EDC	Not specified	Not specified	Not specified
[13]	Electrical stimulation	0–15,000 μA with periodic increments of 30 μA	Peroneal nerve	1—6.6 ± 1.3 mV; 2—4.7 ± 0.8 mV; 3—3.1 ± 0.6 mV; 4—2.3 ± 0.7 mV	Not specified	1—289.0 ± 43.3 mN, 2—257.7 ± 49.1 mN, 3—198.8 ± 71.7 mN, 4—116.4 ± 31.0 mN
[1]	Mechanical stimulation (painful stimulus)	Not specified	Peroneal nerve	Not specified	Not specified	Not specified
[31]	Electrical stimulation	Not specified	Peroneal nerve	3.28 mV ± 0.49 mV. (CNAP 119.47 μV ± 14.87 μV)	0.8–1.55 ms	2451 ± 115 mN en RPNI y 2497 ± 122 mN in control
[3]	Mechanical stimulation (finger movements)	Not specified	FDP, FDS, and EDC	Not specified	Not specified	Not specified
[32]	Electrical stimulation	5–505 μA	Peroneal nerve	1—21.6 ± 9.7 mV. 2—14. ± 6.5 mV	1—3.21 ± 0.53 ms. 2—3.56 ± 0.53 ms	Not specified
[9]	Electrical stimulation	Until reaching the maximum CMAP	Peroneal nerve	1—3.52–6.05 mV, 2—5.3–8.19 mV, 3—10.18–11.59 mV, 4—10.5–11.33 mV	Not specified	Not specified
[33]	Mechanical stimulation (treadmill)	Not specified	Not specified	Not specified	Not specified	Not specified
[34]	Mechanical stimulation (monofilament)	Not specified	Not specified	Not specified	Not specified	Not specified
[35]	Electrical stimulation	143.8 μA at 3 months and 99.6 μA at 4 months	Sural nerve	0.68 mV, at 3 months and 2.27 mV at 4 months	Similar to control group	Not specified
[36]	Electrical stimulation	1—140 ± 50 μA, 2—51 ± 20 μA	Peroneal nerve	1—19.4 ± 4.8 mV, 2—23.4 ± 11.9 mV	1—1.21 ± 0.16 ms, 2—1.2 ± 0.16 ms	Not specified
[10]	Electrical stimulation	Until reaching the maximum CMAP	Peroneal and tibial nerve	Not specified	Not specified	Not specified
[37]	Not specified	Not specified	Not specified	Not specified	Not specified	Not specified
[38]	Not specified	Not specified	Not specified	Not specified	Not specified	Not specified
[39]	Electrical stimulation	Until reaching the maximum CMAP	Peroneal nerve	Not specified	Not specified	Not specified
[40]	Electrical stimulation	200–3000 μA	Sciatic nerve	50–500 μV	Not specified	Not specified
[41]	Not specified	Not specified	Not specified	Not specified	Not specified	Not specified
[63]	Mechanical stimulation (finger movements)	Not specified	Median nerve	0.4–0.6 mV	Not specified	Not specified
[42]	Electrical stimulation	Not specified	Peroneal nerve	1—6.7 mV; 2—5–1.16 mV	Not specified	1—500 mN; 2—5–137 mN
[43]	Electrical stimulation	Not specified	Tibial nerve	1—5.8 ± 3.82 mV; 2—1.4 ± 0.9 mV	1—1.9 ± 0.49 ms; 2—2.2 ± 0.66 ms	Not specified
[45]	Electrical stimulation	Not specified	Peroneal nerve	1—24.2 ± 9.4 mV; 2—6.8 ± 7.1 mV	Not specified	1—2658 ± 558 mN; 2—1627 ± 493 mN
[44]	Electrical stimulation	Not specified	Not specified	Not specified	Not specified	Not specified
[46]	Not specified	Not specified	Not specified	Not specified	Not specified	Not specified
[47]	Not specified	Not specified	Not specified	Not specified	Not specified	Not specified
[48]	Not specified	Not specified	Not specified	Not specified	Not specified	Not specified

## 4. Discussion

The University of Michigan’s prominent presence in twenty-eight out of the thirty-five reviewed articles underscores its leadership in researching the RPNI technique in animal models. We hope that other institutions will validate and advance RPNI applications in the near future.

Given that performing RPNIs is not technically difficult, we anticipate an increase in the utilization of animal models and RPNI applications for pain management in humans. A longer path is expected in the case of the RPNI and myoelectric prosthesis, as it entails the need for more extensive technical resources, including the prosthetic device itself and the connection between the RPNI and the prosthesis.

### 4.1. Type of Model

The predominance of rat models in thirty-two instances is a common practice due to their widespread use in biomedical research. Rhesus macaques, while representing a different order of magnitude in complexity, were utilized in only three studies [3,62,63]. While rats provide practical advantages, such as cost-effectiveness and ease of handling, the translation of findings to larger primates and ultimately to humans may face challenges given the considerable biological differences.

The distinction between Inlay-RPNI and Burrito-RPNI designs provides valuable insights into the diversity of methodologies. We have observed that the Inlay-RPNI design is recently more often used in published articles than the Burrito-RPNI design. A recent study has demonstrated that the Inlay-RPNI model yielded superior outcomes in preventing neuromas compared to the Burrito-RPNI [48]. In the near future, we anticipate the identification of the most suitable RPNI model for specific applications.

The preference for the common peroneal nerve and EDL muscle in RPNI construction is consistent across studies. We believe that the nerve and muscle selection is not so critical to reproduce the RPNI model. However, the combination of two nerves from the same extremity (such as peroneal and tibial nerve [10,27,28]) may induce difficulties in carrying out some basic activities, such as walking or feeding.

It is worth noting that the RPNI technique has been assessed in various clinical studies involving humans, showcasing promising outcomes in alleviating neuropathic pain and in the application of myoelectric prostheses. Nevertheless, future researchers should prioritize addressing the dearth of clinical trials that substantiate these findings [12,50,54,57,58,64,65].

### 4.2. Aim of the Study

We have categorized our analysis into four distinct aims based on the research question. This offers a structured approach to understanding the multifaceted aspects of RPNI applications. The identified aims (neuroma prevention, myoelectric prostheses development, histological analysis, and neurophysiological analysis) encompass a broad spectrum of RPNI applications, demonstrating the versatility and potential of this technique.

We think the use of the RPNI model for myoelectric prostheses could be the most promising application. Being able to obtain the information from different peripheral nerve fascicles could be a paradigm change in peripheral nerve surgery. However, we identified only six studies focusing mainly on myoelectric prosthesis development [3,10,11,34,62,63]. One of the main limitations of RPNIs is the difficulty of getting the electrical signal from the muscle to the prosthesis. As we have previously summarized, the low amplitude of the electrical signal and the small size of the muscle graft are the main drawbacks. A subcutaneous electromyographic recording could facilitate the acquisition, amplification, and transmission of the electrical signal from the RPNI to the prosthesis.

### 4.3. Histological Analysis

One of the main concerns when analyzing RPNIs is the blood supply of the muscle graft. Muscle tissue is known for its high demanding oxygen requirements [66,67]. The combination of these high muscle metabolic rates and the absence of an established vascular system may hinder efficient oxygen delivery, elevating the risk of complications, like necrosis. Muscular neoangiogenesis, tissue viability, axonal regeneration, or neuroma and fibrosis formation has been evaluated in eighteen publications. No necrosis or muscle graft failure was reported in the animal series analyzed in this review. However, the size of the muscle graft has been associated with the above parameters. Muscle grafts mass exceeding 300 mg presented with worse tissue viability and higher rates of complications, such us fibrosis [13].

Vascularized RPNIs (using vascularized muscle, but not a muscle graft) have been reported, and promising results focusing on neuropathic pain have been published [56,68,69,70]. Despite previous studies analyzing the vascularization of muscle grafts in standard RPNIs, we believe that vascularized RPNIs should yield more stable results. Vascularization of the RPNI is one of our primary concerns, particularly when considering the potential use of a needle to obtain an electric signal from the muscle graft.

One notable limitation within the examined studies lies in the deficiency of objectivity in the histological analysis of samples across diverse research investigations. This shortfall is attributed to the absence of standardized criteria that would facilitate the comparison of histological findings across these studies. The inclusion of subjective terms, such as “good”, “viable”, or “healthy” introduces inherent ambiguity, thereby impeding the ability to conduct comprehensive comparative analyses among the various studies. Addressing this limitation necessitates the establishment of clear and standardized criteria, which is crucial for promoting objectivity and enhancing the reliability of histological assessments in future research endeavors.

### 4.4. Neurophysiological Analysis

The inclusion of both electrical and mechanical stimulation (monofilaments, treadmill, and nociceptive tactile stimuli) in the studies contributes to a comprehensive understanding of RPNI outcomes. One of our concerns is that the maximum muscle strength [1,3,9,10,11,15,23,25,26,27,28,30,32,33,34,35,36,37,38,39,40,41,43,44,46,47,48,62,63] and the stimulus intensity [1,3,9,10,11,15,24,25,26,27,28,29,31,33,34,37,38,39,40,41,42,43,44,45,46,47,48,63] were not reported in some studies. Given the relatively short distance between the stimulation site and the RPNI, artefacts could potentially mask the proper reinnervation of the RPNI. We believe that receiving an electrical signal in such a small muscle is one of the major challenges in handling and transmitting that signal to a myoelectric prosthesis. Future studies will need to demonstrate how to optimize this signal and reliably capture the action potential despite electrical noise.

### 4.5. Limitations and Future Challenges

We believe that this systematic review will be very useful in aiding future researchers to enhance surgical techniques and the application of RPNIs across various animal models. Given the significant technical complexity involved in using RPNIs for electrical signal acquisition and myoelectric prosthesis control, we are convinced that refining the animal model of RPNIs could directly impact its application in human contexts. This advancement may signify a significant step towards optimizing procedures and the future viability of RPNIs in clinical applications.

The use of RPNI involves substantial challenges in its clinical implementation. Tissue viability (given its nature as non-vascularized muscle grafts) or limitations in detecting and amplifying electrical signals in RPNI directly impact their functional effectiveness. Transitioning RPNI models from an animal to a human setting presents potential obstacles, given the potential influence of physiological variations on their effectiveness and response. Finally, configuring and adapting patients to prostheses derived from RPNI pose challenges in terms of acceptance and optimal functionality in daily life. These aspects underscore the complexity and potential barriers to be addressed during the development and implementation of RPNI and their clinical applications.

## 5. Conclusions

To the best of our knowledge, this is the first systematic review of the RPNI technique in animal models. Murine models of RPNIs have consistently demonstrated promising results among several studies, particularly in the myoelectric prosthetics field and the prevention of neuropathic pain. Histological, neurophysiological, and functional analyses are summarized to be used in further studies. Forthcoming research should aim to validate these findings and continue to improve the synergy between humans and machines, advancing a more sophisticated interaction paradigm.

## Data Availability

Not applicable.

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
