# Peer review of "Regenerative Peripheral Nerve Interfaces (RPNIs) in Animal Models and Their Applications: A Systematic Review"

_ijms, 2024, doi:10.3390/ijms25021141_

Round 1
Reviewer 1 Report
Comments and Suggestions for Authors
This study systematically reviewed current research on using regenerative peripheral nerve interfaces in animal models. It provides innovative findings for its potential audience. Several suggestions are listed below to improve this manuscript further.
- The title needed to be better structured, as the word “applications” following “the use” may lead to confusion.
- Please provide references for the third paragraph of the Introduction.
- Detailed reasons for exclusion and numbers for enrolled literature in each step need to be listed in Figure 1.
- Please prepare Tables 1-4 based on the guidelines on formatting tables for publication.
- It is difficult to follow when using No to identify enrolled studies. Please re-list enrolled studies (n=35) with the specific reference No in Tables 1-4.
- In Table 3, please add group details for the histology results in each study. What do you mean by stating “good” for the histological results? Please specify.
- The quality of enrolled research should have been assessed in the Results. Further, the limitations of enrolled studies and the problems of the focused topic should have been addressed in the Discussion. Please revise.
- The translational value of this study needs to be further discussed.
- Please provide more details about the data collection process, such as the methods used and how many reviewers were involved.
Comments on the Quality of English LanguageMinor edits to improve the clarity are desired.
Reviewer 2 Report
Comments and Suggestions for Authors
The authors present a systematic review on the regenerative peripheral nerve interface technique in animal models. They summarize the literature concisely and offer their insights. The manuscript might be improved by considering the following comments and questions.
Results: Line 80: Please change “Michigan University” to “The University of Michigan.”
Table 1: For the column labeled “university,” you do not have to write “university” in every cell. “Florida University” does not really exist. We have the University of Florida, Florida State University, Florida International University, Florida Atlantic University, all of which are different institutions.
Discussion: 3.3. Aim of the study: Paragraph 2: RPNI has been around for many years, but I have not heard of any clinical trials. What are the biggest limitations to clinical translation? Please elaborate on the next research steps needed to move this towards the clinical arena.
